# Revealing the Coupling Relationship between the Gross Ecosystem Product and Economic Growth: A Case Study of Hubei Province

**Shuai Guan [1], Qi Liao [2,3], Wenjun Wu [4,5,6], Chuan Yi [3] and Yueming Gao [4,5,*]**

[1] School of Public Administration, Central China Normal University, Wuhan 430079, China; guanshuai@mails.ccnu.edu.cn

[2] College of Resources & Environment, Huazhong Agricultural University, Wuhan 430070, China; liaoqi@hbaes.ac.cn

[3] Hubei Provincial Academy of Eco-Environmental Sciences, Wuhan 430072, China; yichuan@hbaes.ac.cn

[4] State Environmental Protection Key Laboratory of Environmental Planning and Policy Simulation, Chinese Academy of Environmental Planning, Beijing 100012, China; wuwj@caep.org.cn

[5] The Center for Eco-Environmental Accounting, Chinese Academy of Environmental Planning, Beijing 100012, China

[6] The Innovation Center for Eco-environment-Oriented Development, Chinese Academy of Environmental Planning, Beijing 100012, China

* Correspondence: gaoym@caep.org.cn

**Abstract:** The question of how to balance rapid economic growth with ecosystem pressures has become a key issue in recent years. Using the Tapio decoupling model, the spatial autocorrelation model, and the LMDI decomposition model, we analyzed the spatiotemporal variation in gross ecosystem product (GEP) in Hubei Province, studied the relationship between GEP and economic growth, and analyzed the driving factors of GEP variation. The results show that, during the period 2010–2019, the decoupling coefficient between GEP and economic growth in Hubei Province gradually decreased, while the decoupling relationship changed from weak decoupling to strong decoupling; this change is reflected not only in the decoupling index values of various cities but also in the number of changing cities, so this negative change should attract the attention of policy-makers. In addition, there is a significant local spatial autocorrelation in Hubei Province, mainly distributed in the northwest and southwest of the province, and the trend is becoming increasingly obvious. As the decoupling trend is negative, it is necessary to pay attention to local autocorrelation changes, especially in highly correlated cities, and take action to prevent the further exacerbation of such decoupling to maintain healthy economic and social development. Regarding the driving factors of GEP changes in Hubei Province, cities with strong decoupling and those with weak decoupling have certain differences, and different types of decoupling cities need to adopt different strategies to alleviate pressure on the ecological environment. Cities with a weak decoupling need to address the problem of pollutant emissions associated with industrial upgrading and the positive impact of scientific and technological innovation on the ecological environment. Cities with strong decoupling should not only address pollutant discharge but also improve the area of ecological land. From the perspective of urban development, the high-quality development trend of Wuhan, Yichang, Xiantao, Qianjiang, Xianning, and other cities shows a continuous trend of improvement. Ezhou, Jingzhou, Shennongjia, and other cities need to guard against the loss of ecosystems caused by economic growth.

**Keywords:** GEP; ecosystem services; Hubei Province; decoupling relationship; GDP

## 1. Introduction

The world is facing increasingly serious global problems, such as resource depletion and ecological degradation, which makes people reflect on the need to change the economic

growth model at the cost of destroying the ecosystem in the past. China's rapid economic growth over the past four decades may come at the cost of destroying the ecosystem. It is worth noting that China has taken note of this issue and incorporated "high-quality development" into the country's key development policies to seek a green economic growth model that promotes both economic growth and ecosystem development; however, it has not been concluded whether this development model has successfully achieved the set goals, and therefore, research on the relationship between China's economic growth and ecosystem has attracted extensive attention [1,2].

Ecosystem service value (ESV) is an important indicator to evaluate regional ecosystem quality. Scholars have calculated ESV at different scales, such as global, national, regional and watershed. Among them, Costanza estimated the global ESV and found that the average value of the entire biosphere is USD 33 trillion per year [3]. Chen et al. plotted the distribution of ESV in China and calculated that the total value of ESV is CNY 7.78 trillion per year [4]. Jiang et al. found that the ESV of the Qinghai-Tibet Plateau has increased from USD 1.77 trillion in 1990 to USD 1.81 trillion in 2015 [5]. Bai et al. calculated the ESV for the Manas River Basin in 2016 at USD 2.6 billion [6]; these studies raise public awareness of ESV and stimulate public actions to protect the ecosystem, yet they ignore the possibility of using ESV to guide socioeconomic development policy-making [7]. In China, scholars focus on the development of regional gross ecosystem value (GEP). GEP is a monetary value indicator that measures the total value of the final goods and services that the ecosystem provides for human well-being and socioeconomic development [8]. In other words, GEP can be used to give people a clear idea of the ESV through a simple monetary value, and it can more accurately reflect the physical geography and socioeconomic characteristics of the region [9,10].

Scholars have made achievements in studying economic growth and ecosystem by using decoupling analysis. Qian et al. investigated the decoupling relationship between sulfur dioxide and the industrial economy in China from 1996 to 2015, and found that most provinces with weak decoupling states are located in less developed provinces with energy-intensive industries [11]. Zhang et al. analyzed the decoupling relationship between the overall water environmental pressure and economic growth in the Yangtze River Economic Belt and found that there is a trend from weak decoupling to strong decoupling [12]. Shuai et al. studied the relationship between economic growth and carbon emissions in 133 countries and found that a higher income-level group has a larger proportion of countries having reached their decoupling status [13]. Shan et al. studied the relationship between urban economic growth and carbon emissions in China through decoupling analysis and driver factor analysis and found that improvement in production and carbon efficiency is the most important driver [14]. Liang et al. studied the relationship and driving factors of decoupling between economic growth and environmental pressure in China and found that China has achieved relative decoupling in general, and found the most important driving contributors from the perspectives of producers and consumers [15]; however, research on the decoupling analysis mostly focuses on economic growth and energy consumption, environmental pressure and carbon emissions, few studies have combined the ESV with and socioeconomic development, which could provide another important perspective on protecting ecosystems. The ecosystem and economy are the two systems most closely related to human beings and socioeconomic development. Therefore, it is of great significance to study whether there is a direct decoupling relationship between them in the development process, what is the evolution law of this decoupling relationship, and what factors drive the changes in regions with different types of decoupling. To this end, we take Hubei Province as a case to explore the relationship between GEP and economic growth from the perspective of decoupling and driving factors.

The relevant estimation methods and data sources can be found in Section 2 of this paper, which are the GEP measurement method, the Tapio decoupling method, the spatial autocorrelation model and the LMDI decomposition method. The decoupling relationship between GEP and economic growth, the spatial distribution characteristics of this decou-

pling relationship and the driving factors of GEP changes in cities with different decoupling types are presented in Section 3. We further discuss the relationship between GEP and regional economic growth and socioeconomic development in Hubei province, and point out the shortcomings in Section 4. The conclusions of the paper are presented in Section 5.

## 2. Methods and Data Sources

### 2.1. GEP Model

The GEP is the economic value of products and services provided by ecosystems for human survival and well-being, while ecosystem products and services refer to the material resources and conditions provided by ecosystems and ecological processes for human survival, production, and life. Measuring ecosystem services can help in the analysis and evaluation of their economic value for human well-being [16,17]. On the basis of summarizing the United Nations SEEA, MA, Chinese Technical Guide for the Accounting of Terrestrial GEP, as well as the relevant achievements and experiences of well-known scholars [3,18–21], we take into consideration the availability of data and propose a method for measuring GEP in Hubei Province, as shown in Formula (1) and Table 1; this can be calculated from the three perspectives of an ecosystem provisioning service, ecosystem regulating service, and ecosystem ecotourism service.

$$GEP = EPV + ERV + ECV \tag{1}$$

where EPV represents the monetary value of the ecosystem provisioning service, ERV represents the monetary value of the ecosystem regulating service, and ECV represents the monetary value of the ecosystem ecotourism service. The specific categories involved are shown in Table 1.

**Table 1.** The methods used for accounting GEP.

| Primary Indicators | Secondary Indicators | Monetary Value |
|---|---|---|
| EPV | Production of ecosystem goods | Market value method |
| | Water supplement | Market value method |
| ERV | Soil retention | Replacement cost method |
| | Nonpoint pollution control | Replacement cost method |
| | Climate regulation | Replacement cost method |
| | Carbon sequestration | Market value method |
| | Oxygen release | Market value method |
| | Flood mitigation | Replacement cost method |
| | Water retention | Replacement cost method |
| | Air purification | Abatement cost method |
| | Water purification | Replacement cost method |
| | Peat control | Market value method |
| ECV | Ecotourism | Travel cost method |

### 2.2. Decoupling Analysis

Decoupling theory refers to the basic theory of blocking the connection between economic growth and resource consumption or environmental pollution. At the end of the 20th century, the OECD introduced the concept of decoupling into agricultural policy research and gradually expanded it to environmental analysis and other fields. The Tapio decoupling model is suitable for analyzing the correlation between two types of indicators. For example, this model is used to study the relationship between carbon dioxide emissions and the development of industries [22–24], the relationship between urban economic growth and solid waste generation [25], and the relationship between economic growth and energy consumption [13,26].

Changes in GEP and GDP can be used to study the decoupling relationship between ecosystem service value and economic growth in Hubei Province. Tapio proposed that decoupling indicators be defined as follows:

$$e = \frac{\Delta \text{GEP}/\text{GEP}^0}{\Delta \text{GDP}/\text{GDP}^0} \tag{2}$$

where $e$ represents the decoupling elasticity coefficient, $\Delta \text{GEP}/\text{GEP}^0$ represents the growth rate of GEP, and $\Delta \text{GDP}/\text{GDP}^0$ represents the growth rate of GDP.

To distinguish different decoupling conditions, we divide the calculated results into eight types, as shown in Table 2.

**Table 2.** Types of decoupling.

|  | Types | Features |
|---|---|---|
| Economic growth | Expansive negative decoupling | $\Delta \text{GEP} > 0, \Delta \text{GDP} > 0, e > 1.2$ |
|  | Expansive connection | $\Delta \text{GEP} > 0, \Delta \text{GDP} > 0, 0.8 \leq e \leq 1.2$ |
|  | Weak decoupling | $\Delta \text{GEP} > 0, \Delta \text{GDP} > 0, 0 \leq e < 0.8$ |
|  | Strong decoupling | $\Delta \text{GEP} < 0, \Delta \text{GDP} > 0, e < 0$ |
| Economic recession | Declining decoupling | $\Delta \text{GEP} < 0, \Delta \text{GDP} < 0, e > 1.2$ |
|  | Declining connection | $\Delta \text{GEP} < 0, \Delta \text{GDP} < 0, 0.8 \leq e \leq 1.2$ |
|  | Weak negative decoupling | $\Delta \text{GEP} < 0, \Delta \text{GDP} < 0, 0 \leq e < 0.8$ |
|  | Strong negative decoupling | $\Delta \text{GEP} > 0, \Delta \text{GDP} < 0, e < 0$ |

### 2.3. Spatial Autocorrelation Analysis

Spatial correlation analysis represents the correlation between a cell and its surrounding cells in a region, including single-variable spatial correlation analysis (global and local) and double-variable spatial correlation analysis. Global spatial autocorrelation describes the spatial features of attributes on the whole scale through Moran's *I*, while local spatial autocorrelation reflects the relation between a given cell and the adjacent cells for a particular attribute through a LISA cluster diagram. Double-variable spatial correlation analysis reveals the correlation of two variables through a LISA cluster diagram. The dependency and heterogeneity between the urbanization index and ecosystem services are characterized using the spatial analysis tools in the GeoDa software, and their spatial correlation is measured with Moran's *I*. Since spatial autocorrelation analysis can judge the correlation between adjacent areas, it has a wide range of applications; thus, it has been used by scholars for many tasks, such as assessing housing price changes [27], plant distribution [28], animal distribution [29], and cancer mortality [30].

The specific calculation formula is as follows:

$$I = \frac{n \sum_{i=1}^{n} \sum_{j=1}^{n} W_{ij} \left( X_i - \overline{X} \right) \left( X_j - \overline{X} \right)}{\left( \sum_{i=1}^{n} \sum_{j=1}^{n} W_{ij} \right) \sum_{j=1}^{n} \left( X_i - \overline{X} \right)^2} \tag{3}$$

where $n$ represents the number of spatial units, $X_i$ and $X_j$ are the values of variable X in adjacent matching spatial units, $\overline{X}$ is the average value of the attribute in $n$, and $W_{ij}$ is an element of the binary spatial weight matrix $W$ constructed by first-order rook contiguity based on a common boundary. If $W_{ij} = 0$, then space units $i$ and $j$ are not neighbors, while $W_{ij} = 1$ indicates that they are adjacent.

According to Formula (3), $I$ has a value between $-1$ and 1, where $I > 0$ indicates a positive correlation, $I < 0$ indicates a negative correlation, and $I = 0$ indicates that there is no correlation. The greater the absolute value of $I$ is, the stronger the correlation between spatial units will be, which means that the spatial agglomeration level is higher. The closer the absolute value of $I$ is to 0, the closer the spatial distribution will be to an independent random distribution.

### 2.4. Driving Factor Analysis

The LMDI decomposition method is based on mathematical calculations, which are used to decompose the target value into several factors, determine the degree of contribution of other factors, and determine the factors that have the greatest influence on the target value [31]. GEP is a measure of the value of ecosystem services, and both natural and socioeconomic factors can change the GEP. In this paper, the GEP is decomposed into seven driving factors. The decomposition formula and driving factors are as follows:

$$\text{GEP} = \frac{E}{C} \times \frac{C}{R} \times \frac{L}{G} \times \frac{G}{S} \times \frac{S}{H} \times \frac{H}{L} \times R \tag{4}$$

$$\text{GEP} = \text{EC} \times \text{CR} \times \text{LG} \times \text{GS} \times \text{SH} \times \text{HL} \times R \tag{5}$$

where E stands for the GEP ($10^8$ CNY), C stands for the cultivated area (hectare), R stands for the proportion of the rural population (%), G stands for the GDP ($10^8$ CNY), L stands for the local public financial expenditure ($10^8$ CNY), H stands for the added value of a high and new technology industry ($10^8$ CNY), and S stands for industrial sulfur dioxide emissions (ton).

EC represents the GEP value per unit of cultivated land area. The larger the value is, the greater the GEP per unit area of cultivated land will be. CR represents access to arable land for the agricultural population, and LG represents the fiscal expenditure of local governments. The higher the value is, the more the local government is likely to spend on ecological maintenance. GS represents the pollution factors due to economic development; the higher this value is, the fewer pollutants are discharged and the more ecological environment benefits there are. SH represents the pollution of the high-tech industry; the higher the value is, the more serious the pollution is. HL represents the development of the local high-tech industry; the higher the value is, the more dependent the local government is on the high-tech industry.

To explore the effect of the seven factors on the change in GEP, we decomposed GEP according to the following formula [32]:

$$\Delta \alpha = \alpha^t - \alpha^0 = \sum_{i=1}^{7} \Delta \alpha_i \tag{6}$$

$$\Delta \alpha_i = \varepsilon \times \ln\left(\frac{\alpha^{it}}{\alpha^{i0}}\right) \tag{7}$$

$$\varepsilon = \frac{\alpha^t - \alpha^0}{\ln \alpha^t - \ln \alpha^0} \tag{8}$$

where $\Delta \alpha$ represents the change in the GEP, $\alpha^t$ represents the GEP in year $t$, $\alpha^0$ represents the GEP in the base year, and $\alpha_i$ represents the seven driving factors.

### 2.5. Data Sources

The EPV and ECV data were sourced from the Hubei Statistical Yearbook (2011–2020) and the statistical yearbooks of 17 cities of Hubei Province (2011–2020); market prices were taken from the websites of the Hubei Provincial Development and Reform Commission (http://fgw.hubei.gov.cn/), the National Agricultural Product Business Information Public Service Platform (https://nc.mofcom.gov.cn/), and China Water Price Net (https://www.h2o-china.com/). The ERV data were derived from the Resource and Environment Science and Data Center (http://www.resdc.cn/); the Institute of Soil Science, CAS (http://www.issas.ac.cn/); USGS (https://lpdaac.usgs.gov/products/myd16a3gfv006/ (accessed on 14 December 2021)); China Hubei Emission Exchange (http://www.hbets.cn/index.php/index-show-tid-15.html (accessed on 14 December 2021)); Institute of Geographic Sciences and Natural Resources Research, CAS (http://www.igsnrr.ac.cn/); Water Resources Bulletin of Hubei Province (2011–2020); and China Forestry and Grassland Statistical Yearbook (2010–2019).

## 3. Results

### 3.1. Spatiotemporal Change Analysis of GEP in Hubei Province

The results (Figure 1) show that GEP remained stable at approximately CNY 7 trillion from 2010 to 2019, with the lowest value recorded in 2010 (CNY 6.78 trillion) and the highest value found in 2015 (CNY 7.22 trillion). Jingzhou was in the first position in terms of GEP value in Hubei Province from 2010 to 2019. Wuhan joined the first stage in 2015, mainly due to the rapid transformation of its ecotourism service value. In addition, Shiyan, Xianning, and Xiaogan were the three prefecture-level cities that showed an obvious growth in GEP, and Xiaogan moved up from the third level to the second level. Figure 1d shows that the GEP of the six cities decreased from 2010 to 2019—namely, in Jingzhou (CNY −54 billion), Huanggang (CNY −31 billion), Huangshi (CNY −23 billion), Shennongjia (CNY −15 billion), Xiangyang (CNY −11 billion), and Ezhou (CNY −9 billion). These led to −30%, −17%, −13%, −8%, −6%, and −5% of the GEP variation in Hubei Province, respectively. The GEP of the 11 cities increased, and the top 5 cities were Wuhan (CNY 115 billion), Xiaogan (CNY 53 billion), Shiyan (CNY 45 billion), Xianning (CNY 33 billion), and Enshi (CNY 22 billion), which contributed 63%, 29%, 25%, 18%, and 12% of the variation in the value of GEP, respectively.

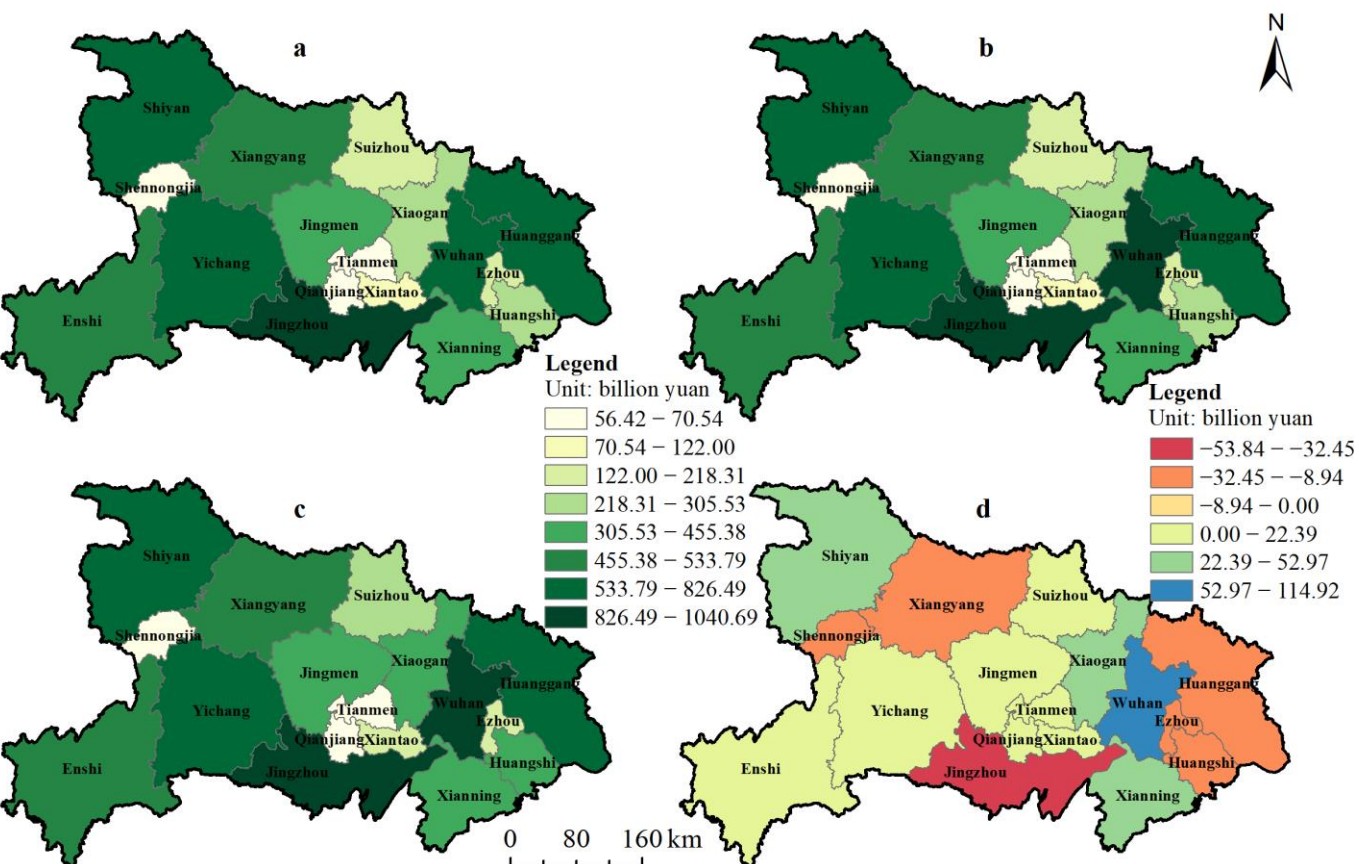

**Figure 1.** The 17 administrative regional distributions of GEP in Hubei Province ((**a**): the GEP in 2010; (**b**): the GEP in 2015; (**c**): the GEP in 2019; (**d**): GEP changes from 2010 to 2019).

### 3.2. Decoupling Analysis of GEP and Economic Growth

3.2.1. Overall Decoupling State Analysis

To clarify the relationship between ecosystem service value and economic growth (Table 3), we used the Tapio decoupling model to study the decoupling relationship between GEP and economic growth in the periods of 2010–2015, 2015–2019 and 2010–2019. According to the total GDP of the 17 cities of Hubei Province, the GDP in 2010, 2015, and 2019 was CNY 1586 billion, CNY 3056 billion, and CNY 4587 billion, respectively, with

increases of 92.69% from 2010 to 2015 and 50.10% from 2015 to 2019; this shows that the economy of Hubei Province has developed rapidly in recent years; however, the GEP from 2010 to 2019 first increased and then decreased, with change values of 6.46% and −1.91%, respectively.

**Table 3.** Decoupling status of cities between GEP and GDP in Hubei Province.

| Administrative Region | 2010–2015 | | 2015–2019 | | 2010–2019 | |
|---|---|---|---|---|---|---|
| | $e$ | Type | $e$ | Type | $e$ | Type |
| Wuhan | 0.148 | Weak decoupling | 0.025 | Weak decoupling | 0.077 | Weak decoupling |
| Huangshi | 0.006 | Weak decoupling | −0.186 | Strong decoupling | −0.049 | Strong decoupling |
| Shiyan | 0.104 | Weak decoupling | −0.007 | Strong decoupling | 0.043 | Weak decoupling |
| Yichang | 0.068 | Weak decoupling | −0.094 | Strong decoupling | 0.011 | Weak decoupling |
| Xiangyang | 0.058 | Weak decoupling | −0.160 | Strong decoupling | −0.010 | Strong decoupling |
| Ezhou | −0.032 | Strong decoupling | −0.039 | Strong decoupling | −0.026 | Strong decoupling |
| Jingmen | 0.054 | Weak decoupling | −0.069 | Strong decoupling | 0.007 | Weak decoupling |
| Xiaogan | 0.075 | Weak decoupling | 0.249 | Weak decoupling | 0.109 | Weak decoupling |
| Jingzhou | 0.048 | Weak decoupling | −0.202 | Strong decoupling | −0.028 | Strong decoupling |
| Huanggang | 0.021 | Weak decoupling | −0.141 | Strong decoupling | −0.028 | Strong decoupling |
| Xianning | 0.081 | Weak decoupling | −0.006 | Strong decoupling | 0.039 | Weak decoupling |
| Suizhou | 0.029 | Weak decoupling | −0.021 | Strong decoupling | 0.009 | Weak decoupling |
| Enshi | 0.073 | Weak decoupling | −0.065 | Strong decoupling | 0.023 | Weak decoupling |
| Xiantao | 0.075 | Weak decoupling | 0.123 | Weak decoupling | 0.070 | Weak decoupling |
| Qianjiang | 0.139 | Weak decoupling | 0.061 | Weak decoupling | 0.088 | Weak decoupling |
| Tianmen | 0.144 | Weak decoupling | −0.041 | Strong decoupling | 0.061 | Weak decoupling |
| Shennongjia | −0.019 | Strong decoupling | −0.451 | Strong decoupling | −0.160 | Strong decoupling |
| Hubei | 0.067 | Weak decoupling | −0.067 | Strong decoupling | 0.014 | Weak decoupling |

The decoupling relationship between the ecosystem service value and economic growth in Hubei Province from 2010 to 2019 showed two stages, with the trend moving from weak decoupling to strong decoupling. The coupling relationship between GEP and GDP in Hubei Province showed weak decoupling from 2010 to 2015, with a value of −0.067. In this period, with rapid economic growth, the value of ecosystem services also increased, which showed relatively coordinated development; however, the coupling relationship between GEP and GDP in Hubei Province showed strong decoupling from 2015 to 2019, with a value of −0.067; this shows that the economic growth rate was fast, but the accompanying decrease in the value of ecosystem services showed relatively uncoordinated development. During the period 2010–2019, the variation in the GDP and GEP was greater than 0, the decoupling value was 0.014, and the decoupling stage was weak decoupling; this shows that the GEP was also on the rise over a long period of time with the rapid economic growth in Hubei Province.

The ecosystem service value and economic growth showed coordinated development, which is a positive development mode. Figure 2 shows that the distribution of decoupling relationships was different in the three periods. The decoupling relationship mainly showed weak decoupling from 2010 to 2015, including 88.23% of the 17 cities, except for Ezhou and Shennongjia. From 2015 to 2019, the decoupling relationships mainly showed strong decoupling. Only Wuhan, Xiaogan, Xiantao, and Qianjiang were weakly decoupled, while the other 13 cities were strongly decoupled. In general, the decoupling stage mainly showed weak decoupling from 2010 to 2019. It is worth noting that through this analysis, we were able to find that the decoupling relationship between different cities in Hubei Province ranged from weak decoupling to strong decoupling; this change reflects the economic growth of Hubei Province, and the relationship between the ecosystem service value gradually appears as a reverse change trend, which is a kind of negative change. For the sake of benign socioeconomic growth in Hubei Province, we should take measures to protect and improve the environment of each administrative region.

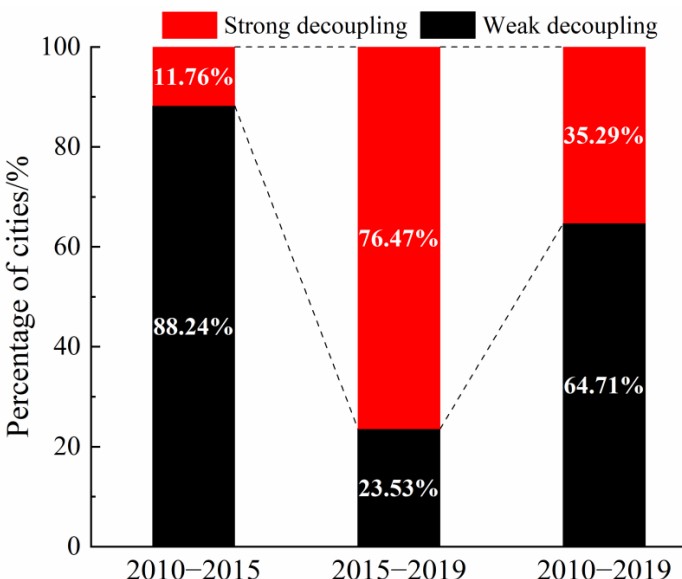

**Figure 2.** Distribution of decoupling stages in Hubei Province.

The economic growth rate was fast, but the accompanying decrease in the value of ecosystem services showed relatively uncoordinated development; this change shows that the relationship between economic growth and the value of ecosystem services changed from relatively coordinated development to relatively uncoordinated development. Economic growth did not bring about any improvement in ecosystem service value, and some cities still showed declining ecosystem service value. Therefore, cities with large changes in their decoupling relationships need to pay special attention to the protection and development of the ecological environment so that the ecosystem service value can be maintained or increased to maintain a good mode of socioeconomic growth.

3.2.2. Spatial Autocorrelation Analysis of GEP and Economic Growth

The calculated results (Table 4) show that the global Moran's I values for Hubei Province in the periods 2010–2015, 2015–2019, and 2010–2019 were −0.009, 0.066, and 0.041, respectively; this means that, in these three periods, the decoupling relationships of Hubei province were spatial negative correlation, spatial positive correlation, and spatial positive correlation, and the overall change trend was transformed from spatial negative correlation to spatial positive correlation; however, the statistical results show that the $p$-value in the three periods was greater than 0.1, indicating that the model was not significant, suggesting that Hubei Province had a weak spatial correlation overall; however, due to the foregoing results, we found that the strong decoupling relationship between Ezhou and Shennongjia gradually expanded to other cities surrounding them. Therefore, in order to explore whether the decoupling relationship in Hubei province has local spatial autocorrelation, we further analyzed the local Moran's I values in Hubei Province.

**Table 4.** Global Moran's I values in Hubei Province.

| Type | 2010–2015 | 2015–2019 | 2010–2019 |
|---|---|---|---|
| GEP | −0.009 (0.300) | 0.066 (0.155) | 0.041 (0.203) |

The values in parentheses are $p$-values.

The results (Figure 3) show that there is a significant local spatial autocorrelation in Hubei Province. The local spatial autocorrelation mainly exists in the northwest and southwest of Hubei Province. From 2010 to 2015, there was local spatial autocorrelation in Xiantao and Jingzhou. The decoupling index value of Jingzhou was low, and the decoupling index value of the surrounding areas was high, indicating a low–high relationship.

The decoupling index value of Xiantao was high, and the decoupling index value of the surrounding area was also high, indicating a high–high relationship. From 2015 to 2019, there was local spatial autocorrelation in the five cities. The decoupling index value of Enshi and Yichang was low, and the decoupling index value of the surrounding regions was also low, indicating a low–low relationship. The decoupling index value of Shiyan was high, while the decoupling index value of the surrounding area was low, indicating a high–low relationship. The decoupling index value of Tianmen and Xiantao was high, and the decoupling index value of the surrounding area was also high, indicating a high–high relationship. From 2010 to 2019, Shiyan and Enshi had a high–low relationship, Yichang had a low–low relationship, Jingzhou had a low–high relationship, and Tianmen and Xiantao had a high–high relationship. In general, there was significant local spatial autocorrelation between economic growth and GEP, and this relationship gradually became significant over the study period. The change in the decoupling relationship of one administrative region will lead to a change in the surrounding areas. Therefore, to maintain a healthy trend of socioeconomic growth in Hubei Province, attention should be paid to the change in local autocorrelation, especially in areas with high–high relationships, and actions should be taken to prevent further increases in such decoupling.

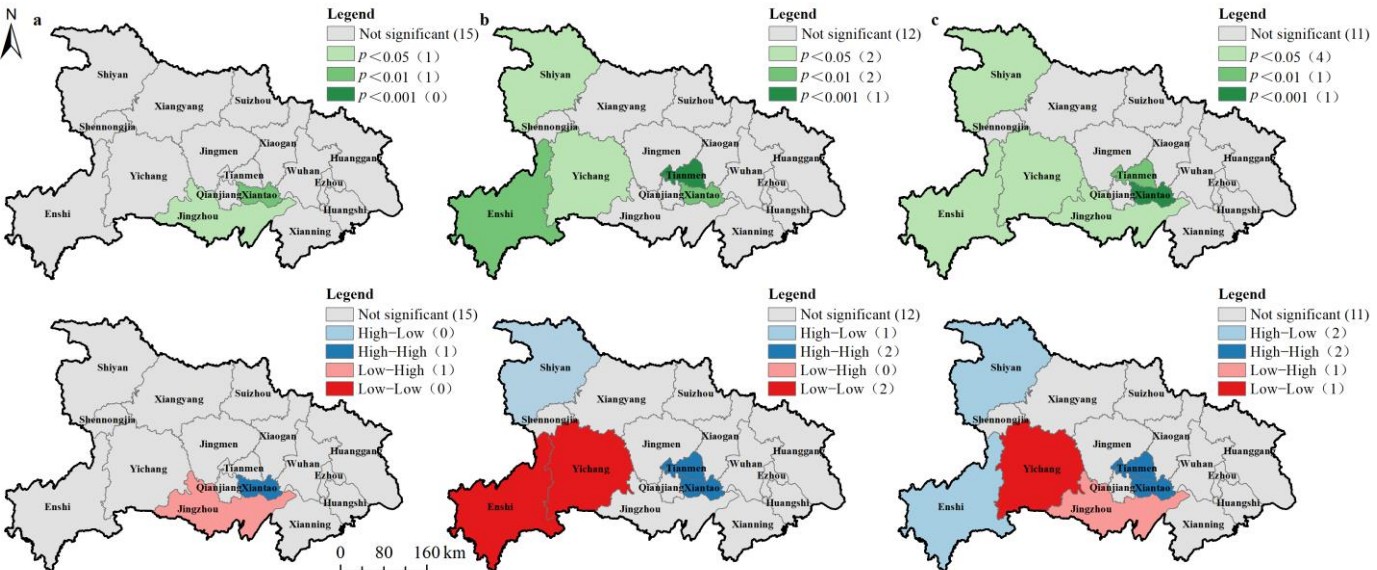

**Figure 3.** Local spatial autocorrelation in Hubei Province (GEP and GDP) ((**a**): 2010–2015; (**b**): 2015–2019; (**c**): 2010–2019. The above figures represent the p values of local Moran's I, and the following figure is the LISA diagram, which was created with the GeoDa software).

### 3.3. Ecosystem Drivers in Terms of the Decoupling Index

After the decomposition of GEP variation in 17 cities of Hubei in the period 2010–2015 and 2015–2019, we obtained some findings that are worth discussing. In cities with strong decoupling (Figure 4a), GEP increased from 2010 to 2015. GS contributed the most to the increase in GEP (accounted for 1932.1%), indicating that with the continuous growth of the regional economy, pollutant emissions will be relatively reduced, thus promoting an increase in the GEP value in the region. Therefore, local governments should introduce large-scale industries with low levels of environmental pollution. LG was also the driver of the increase in GEP (accounted for 536.5%), reflecting that the local government's increased financial investment in environmental protection has a positive effect on the ecosystem. CR and EC were also the drivers of the increase in GEP (accounted for 359.8% and 29.0%), indicating that the increase in the area of land with a high value of ecosystem services represented by cultivated land could improve the value of regional GEP. Therefore, local governments can limit the area of construction land and increase the area of forest, grassland, and cultivated land to improve the value of regional GEP. The driving factors

SH, HL and R had a negative effect on GEP of strong decoupling cities. SH and HL were the two largest negative drivers (accounted for 1495.0% and 973.5%), indicating that with the adjustment of the industry, the emission of pollutants produced by enterprises will reduce the value of GEP in the region. Therefore, the local government should strictly limit the increase in the number and scale of highly polluting enterprises, or reduce pollutant emissions by increasing financial input. R was the driving factor that reduced GEP (accounted for 288.8%), indicating that the rapid growth of agricultural population would reduce the value of regional ecological services. In cities with strong decoupling, GEP decreased from 2015 to 2019. The most important driving factor was SH (accounted for 3019.8%), indicating that pollutant discharge could damage the ecosystem. In cities with weak decoupling (Figure 4b), GEP also increased first and then decreased. The influence of each driving factor was similar to that of cities with strong decoupling, and GS was the most important driving factor to increase GEP, indicating that the growth of regional economy could enhance the ecosystem service value. SH was the most important driving factor to reduce GEP, and SH has a greater driving effect on cities with weak decoupling than cities with strong decoupling, indicating that weak decoupling cities should strictly limit the growth of the number and scale of high-pollution enterprises.

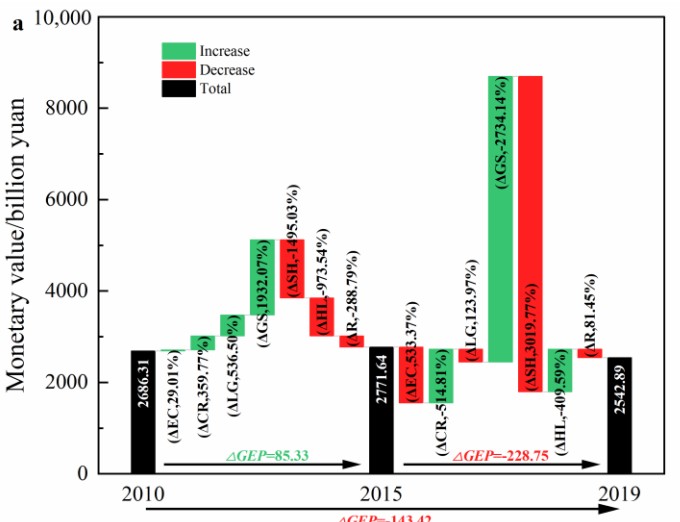 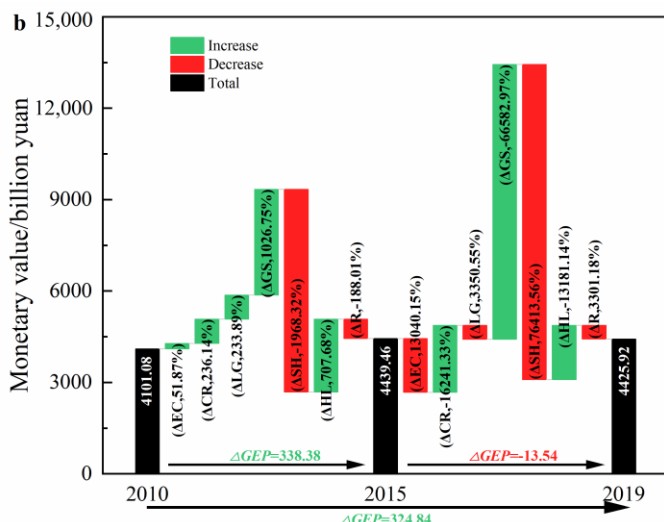

**Figure 4.** Driving factors of GEP in cities with different decoupling levels in Hubei Province ((**a**): strong decoupling cities; (**b**): weak decoupling cities. The pictures show the proportion of changes of each driving factor at different stages, which is the benchmark of ΔGEP).

## 4. Discussion

### 4.1. GEP Works in Parallel with GDP

The Environmental Kuznets curve (EKC) is widely used to explain the relationship between economic development and the ecosystem, which reflects a curving relationship between the ecological environment and economic growth [33,34]. Our results show that, regardless of the strong decoupling cities or weak decoupling cities, with economic growth, GEP in Hubei province from 2010 to 2019 increases first and then decreases, presenting a curving relationship, which reflecting that ecological pressure decreases first and then increases with economic growth. The reason may be that with economic growth, the industrial structure is adjusted to high-pollution enterprises, and the pollutant discharge increases the pressure on the ecological environment [35], thus reducing the ecosystem service value. Our results are consistent with similar other studies [36,37], indicating that there is a curvilinear relationship between economic growth and ecosystem, which can be used as a basis for local governments to make policy adjustments to achieve green economic growth goals.

In addition, we conducted a statical analysis of the relationship between GEP and economic growth in Hubei Province. Taking GEP per unit area and GDP per capita as the intersection, Figure 5 is divided into four quadrants. The cities in Group I should continue to strengthen the role of existing industrial policies in promoting the protection of natural ecosystems and high-quality socioeconomic growth. The cities in Group II should maintain existing social and economic growth, protect the natural ecosystem, and avoid damaging the natural environment. The cities in Group III should maintain a sustainable and eco-friendly development path and explore a path of transformation for ecological products. The cities in Group IV have good natural ecosystem endowments and a high transformation potential for ecological products; these cities should actively support the corresponding industrial structure and implement an ecologically oriented development model. All the cities in Hubei Province showed a trend of high-quality development from 2010 to 2019. In 2010, 64.71% of the cities had the development dilemma of a low GEP and low GDP, and 29.41% of the cities had a development dilemma of a high GEP but low GDP; thus, their ecological value could not be effectively transformed into economic growth. The number of cities in Groups I and II continues to increase, with 5 in 2015 and 11 in 2019. Wuhan has always been located in the first quadrant, and the high-quality development trend is constantly improving. Cities with high-quality development also include Yichang, Xiantao, Qianjiang, and Xianning, which have moved upwards or upwards and to the right over time. Ezhou, Jingzhou, and Shennongjia have exhibited a trend of moving to the upper left, so we need to be alert to their development mode threatening to destroy the ecological environment. Therefore, it is necessary to establish an ecological fiscal transfer payment policy based on GEP, especially for cities with a high GDP and low GEP, to provide ecological compensation to cities with a low GDP and high GEP.

*4.2. The Coupling Relationship between GEP and Socioeconomic Development*

Economic growth promotes the economic and social development of a region and improves people's living standards; however, to measure regional development, we should not only focus on regional economic growth, but also pay more attention to whether it can improve people's living conditions more effectively [38–40]. Most of the existing research focuses on the relationship between economic growth and ecosystem [41,42], little research on the relationship between regional socioeconomic development and ecosystem [43]. Therefore, we further discuss the relationship between these two systems. The HDI (Human Development Index) is an index created by the United Nations Development Programme (UNDP) in 1990 to measure the level of regional development. The HDI measures the three basic indicators of regional life expectancy, education level, and quality of life, which can be used to guide countries and regions to formulate corresponding development strategies [44]. To further discuss the relationship between GEP and regional socioeconomic development, we measured the coupling relationship between GEP and the HDI in 17 cities in Hubei Province.

As shown in Table 5, we found that GEP and regional development, as well as GEP and economic growth, were similar in most cities, with strong decoupling and weak decoupling predominating, and the decoupling status did not change significantly. In weakly decoupled cities, both GEP and HDI had positive changes, but the change in GEP was smaller than that of the HDI, and the development level of these cities was higher than the increase in the ecosystem service value. The GEP of strongly decoupling cities decreased while the HDI increased, indicating that regional development may damage the ecological environment and that local governments should formulate corresponding policies to protect the local ecological environment. From 2010 to 2015, the decoupling status of Wuhan and Qianjiang was an expansive connection, which reflects that the GEP of the two cities was close to the regional development level, and the regional development level and the regional ecosystem could develop in harmony. During the period of 2015–2019, Xiaogan accumulated negative decoupling. During this period, the GEP of Xiaogan in-

creased greatly, but the ecosystem service value failed to effectively improve the regional development level.

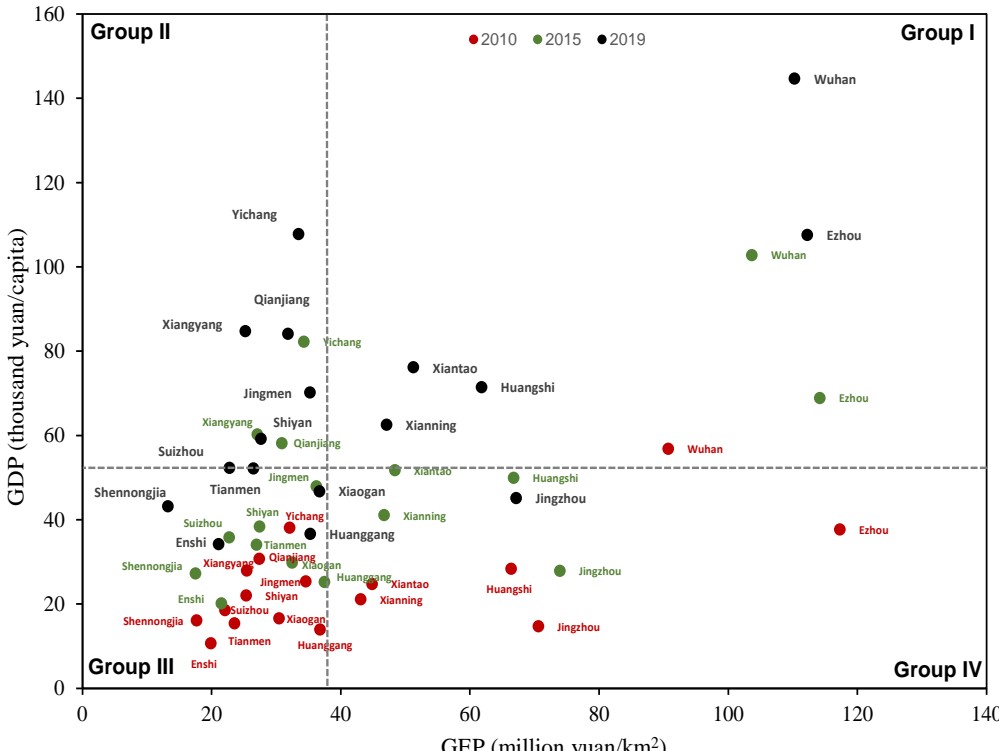

**Figure 5.** Classification of 17 prefecture-level cities on the basis of GEP and GDP from 2010 to 2019. (Group I, located in the right upper quadrant, shows cities both with high GEP and high GDP. Group II, located in the left upper quadrant, shows cities with high GEP but low GDP. Group III, located in the left lower quadrant, shows cities both with low GEP and low GDP. Group IV, located in the right lower quadrant, shows cities with low GEP but high GDP).

**Table 5.** Decoupling status between GEP and HDI for cities in Hubei Province.

| Administrative Region | 2010–2015 | | 2015–2019 | | 2010–2019 | |
|---|---|---|---|---|---|---|
| | $e$ | Type | $e$ | Type | $e$ | Type |
| Wuhan | 0.940 | Expansive connection | 0.156 | Weak decoupling | 0.618 | Weak decoupling |
| Huangshi | 0.028 | Weak decoupling | −0.866 | Strong decoupling | −0.266 | Strong decoupling |
| Shiyan | 0.556 | Weak decoupling | −0.066 | Strong decoupling | 0.353 | Weak decoupling |
| Yichang | 0.308 | Weak decoupling | −0.600 | Strong decoupling | 0.071 | Weak decoupling |
| Xiangyang | 0.331 | Weak decoupling | −1.246 | Strong decoupling | −0.080 | Strong decoupling |
| Ezhou | −0.175 | Strong decoupling | −0.963 | Strong decoupling | −0.267 | Strong decoupling |
| Jingmen | 0.281 | Weak decoupling | −0.298 | Strong decoupling | 0.043 | Weak decoupling |
| Xiaogan | 0.347 | Weak decoupling | 1.571 | Expansive negative decoupling | 0.689 | Weak decoupling |
| Jingzhou | 0.237 | Weak decoupling | −0.776 | Strong decoupling | −0.157 | Strong decoupling |
| Huanggang | 0.071 | Weak decoupling | −0.707 | Strong decoupling | −0.130 | Strong decoupling |
| Xianning | 0.212 | Weak decoupling | −0.031 | Strong decoupling | 0.149 | Weak decoupling |
| Suizhou | 0.081 | Weak decoupling | −0.115 | Strong decoupling | 0.038 | Weak decoupling |
| Enshi | 0.190 | Weak decoupling | −0.294 | Strong decoupling | 0.083 | Weak decoupling |
| Xiantao | 0.152 | Weak decoupling | 0.205 | Weak decoupling | 0.149 | Weak decoupling |
| Qianjiang | 0.814 | Expansive connection | 0.377 | Weak decoupling | 0.657 | Weak decoupling |
| Tianmen | 0.240 | Weak decoupling | −0.878 | Strong decoupling | 0.190 | Weak decoupling |
| Shennongjia | −0.051 | Strong decoupling | −5.352 | Strong decoupling | −0.817 | Strong decoupling |

As shown in Figure 6, the decoupling relationship of 17 cities presented a significant local autocorrelation, which was similar to the decoupling relationship between GEP and

economic growth; however, it is worth noting that the local spatial autocorrelation of two cities differed from the decoupling of GEP and economic growth during the period 2015–2019. The decoupling index of Enshi was high, while that of surrounding areas was low, showing a high–low relationship. The decoupling index of Tianmen was low, while that of the surrounding areas was high, showing a low–high relationship. In general, there was a significant local spatial autocorrelation between GEP and the regional development level, which gradually became significant throughout the study period and was similar to the decoupling that took place between GEP and economic growth. Cities in the southwest and northeast of Hubei province, such as Enshi, Yichang, Jingzhou, and Shiyan, should pay special attention to the spatial autocorrelation and formulate corresponding policies to prevent the local ecosystem from being damaged by their own development or that of the surrounding cities, thus reducing the service value of the ecosystem.

### 4.3. Uncertainty Statement

Values of ecosystem services hugely differ across communities. The benefits of the services are very vague and difficult to understand. The economic value of ecosystem services is subject to manipulation by the interests, knowledge and methods of the researchers. The shortcomings of this study are as follows: First, limited by the availability of statistical and spatial data, this study only calculated the GEP in 2010, 2015, and 2019. Second, we accounted for only limited ecosystem services such as ecotourism in the case of social services. Third, the 13 ecosystem service indicators used in this study may not be applicable to other ecosystems, such as marine ecosystems and desert ecosystems. Fourth, limited by the research depth and time, we only studied the decoupling relationship of Hubei Province based on the relationship between GEP and GDP. More in-depth research on this topic should be conducted, and the research scope could be further expanded in the future.

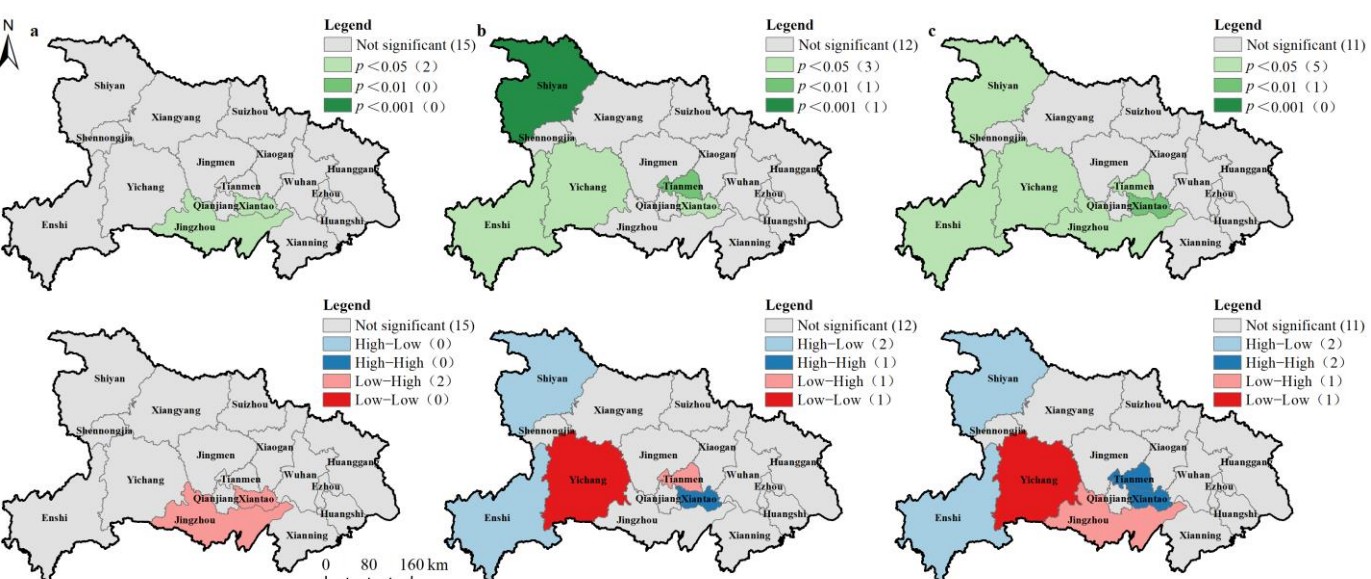

**Figure 6.** Local spatial autocorrelation in Hubei Province (GEP and HDI) ((**a**): 2010–2015; (**b**): 2015–2019; (**c**): 2010–2019).

### 5. Conclusions

This paper using the Tapio decoupling model, the spatial autocorrelation model, and the LMDI decomposition model, analyzed the spatiotemporal variation in GEP in Hubei Province, studied the relationship between GEP and economic growth and analyzed the driving factors of GEP variation. The results show that: first, during the period 2010–2019, the decoupling coefficient between GEP and economic growth in Hubei Province gradually decreased, while the decoupling relationship changed from weak de-coupling to strong

decoupling; this change is reflected not only in the decoupling index values of various cities but also in the number of changing cities, so this negative change should attract the attention of policy-makers. Second, there is a significant local spatial autocorrelation in Hubei Province, mainly distributed in the northwest and southwest of the province, and the trend is becoming increasingly obvious. As the decoupling trend is negative, it is necessary to pay attention to local autocorrelation changes, especially in highly correlated cities, and take action to prevent the further exacerbation of such decoupling to maintain healthy economic and social development. Third, regarding the driving factors of GEP changes in Hubei Province, cities with strong decoupling and those with weak decoupling have certain differences, and different types of decoupling cities need to adopt different strategies to alleviate pressure on the ecological environment. Cities with weak decoupling need to address the problem of pollutant emissions associated with industrial upgrading and the positive impact of scientific and technological innovation on the ecological environment. Cities with strong decoupling should not only address pollutant discharge but also improve the area of ecological land. Finally, it is necessary to establish an ecological fiscal transfer payment policy based on GEP, especially for cities with a high GDP and low GEP, to provide ecological compensation to cities with a low GDP and high GEP.

**Author Contributions:** Writing, Editing, and Mathematical Calculating, S.G., Q.L. and C.Y.; Conceptualization, W.W.; Methodology, Y.G. All authors have read and agreed to the published version of the manuscript.

**Funding:** This research was funded by National Natural Science Foundation of China (Major Program, Grant No. 91846301, "Big data driven public management decision-making innovation model and integration platform"), and Fund project for distinguished young scholars "Research on technical system of ecological environment planning" (40050720).

**Institutional Review Board Statement:** Not applicable.

**Informed Consent Statement:** Not applicable.

**Acknowledgments:** We thank Zhenyu Ding for providing great ideas for our study.

**Conflicts of Interest:** The authors declare no conflict of interest.

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
