# Peer review of "Revealing the Coupling Relationship between the Gross Ecosystem Product and Economic Growth: A Case Study of Hubei Province"

_sustainability, doi:10.3390/su14137546_

Round 1

Reviewer 1 Report

The paper's introduction section is not well organized therefore not well written at all. Too much reference has been cited in the introduction section which makes the reading uninteresting. There is hardly any linking conjunction, adverbs, propositions used in the whole manuscript which makes the paper incoherent. It seems that some references have been picked up from difference sources and jumbled up together. It's also obsolete to use past tenses while stating a reference. There are no concluding sentences at the end of each paragraph of the introduction section. So, the introduction must be rewritten. The proof reading has not been done professionally. A lot of grammatical mistakes are also evident.

The authors must clearly state why this study is important to conduct in the introduction section. The objective of the study is not mentioned in the introduction. The contribution of the study is very poorly stated in two sentences and shockingly understated and disorganized.

It is utterly confusing the purpose of the literature mentioned in the introduction. Are they to mention the shortcomings of the existing literature or for mentioning the importance of conducting this study? The gap or shortcomings of the literature has not been mentioned anywhere in the literature review section. The authors continue to describe different studies but did not mention the gap in the literature. This section must be rewritten as well. It's really hard to understand what authors mean by mentioning these studies. The result and discussion section are very much disorganized. The authors should clearly state which studies are in line with their findings and which contradicts. The paper does not mention any theoretical or anecdotal evidences which actually justify the selection of the variables in the context of the China. Finally the authors need to resubmit the revised paper the the used data with the do file of STATA to verify the examined results.

Reviewer 2 Report

Manuscript: Revealing the Coupling Relationship between Gross Ecosystem Product and Economic Growth: A Case Study of Hubei Province

The study attempted to deepen relationship between economic growth and ecosystem enrichment. The manuscript is worth of publishing with some improvement. I found the following rooms for improvements.

Theoretically the relationship is expected to be inverted U shape. It is because the GDP could grow with efficient management/ uses of the resources. These kinds of theories at opening sections would increase interest of readers.

Compering economic development (economic development indicators including human development index and life span etc) instead of economic growth (GDP) would be more meaningful comparison with gross ecosystems products.

Values of ecosystems services hugely differs with communities. The benefits of the services are very vague and complex to understand. I suggest stating precautionary points that the economic value of ecosystems services are subject of manipulation by academicians/ researches by the interests, knowledge and methods. Most of academicians have worked in economic valuation with limited knowledge about process and services of ecosystems. It is often considered a ballpark figure.

You accounted only limited ecosystems services such as ecotourism in case of social services. There are many missing in your accounting. Such common limitations require acknowledgement.

The authors stated "There are relatively few studies on the decoupling between ecosystem service value and economic growth" to justify the study. There may be more weakness in previous study. These are worth to justify this study. In the conclusion part of the literature review section the authors stated "At present, there is less standard system for the valuation of ecosystem services. In this study, we use a  relatively systematic GEP accounting method". Such points are worth of stating there. 

Method 

There are many groups of ecosystems services. This study is based on only three services. It needs justification. 

Some ecosystems services of some areas are natural gifts that human activities make them little change. These services change little overtimes. Therefore they are little worth of comparing with GDP.  How have you addressed these issues in your analysis?

Results

"Fig. 1(a, b and c) shows the GEP of 17 cities in Hubei Province." Such way of writing in the opening paragraph is not academic standard.

The results presented in Figure 2 is complex to understand. How the graph shows the relationships? It requires more meaningful description.

What the global Moran's I value refers?   

The legend of the Figure 4 are in very small fonts and difficult to read. Some of them need bold letter. They requires separating to make the figure readable.

The subtitle "policy analysis is confusing. There could be better subtitle.

Others

There is missing of discussion section where the authors require providing some scholarly arguments to be such phenomena. The findings are also bench marked with other studies in this section. 

Conclusion section is also missing. This is the section that you highlight home take message and draw policy implications. 

Round 2

Reviewer 1 Report

The authors have not incorporated the suggested points in detail. They are requested to revise previous comments in detail. 

Reviewer 2 Report

The authors have reasonably address most of the comments and suggestion. But there is still major problems.

  1. Introduction section is still too long and included trivial information.
  2. Major problem are in the section titled "3.3 Ecosystem drivers in terms of the decoupling index". Many graphs are included in Figure 4. The authors have acknowledge the figure. The information loaded in each graph is itself in complex to understand. It has included many graphs which has  has made the figure too complex. It requires separating the figures and interpreting them individually or comparatively. I do recomend to publish this article as it is.   

Round 3

Reviewer 1 Report

The authors have modified my all comments in detail.

Author Response

Thank you for putting forward many valuable suggestions on our manuscript, which make our manuscript more scientific and rigorous. 

Reviewer 2 Report

Most of issues are improved but there are some areas which need further improvement. Those are as follows

Introduction section:

I suggest deleting the last paragraph of the introduction section. Instead explain the organizations (structure) of manuscript.

Result section

Most graphs require improvement in labels. The figures are difficult to read. You could make them readable by separating and enlarging figures  

Discussion section

Discussion section requires some improvement in structure of language. For example, Line 309-310: “Our results support scholars' findings that there is a curving relationship between the ecological environment and economic growth”, supposed to be “our results are consistent with similar other studies (provide References)”.

Conclusion section

In this section answers to the following questions (stated in introduction section) are mandatory.

“Whether there is a direct decoupling relationship between them in the development process, what is the evolution law of this decoupling relationship, and what factors drive the changes of regions with different types of decoupling”
